# Effect of Graphene Sheets Embedded Carbon Films on the Fretting Wear Behaviors of Orthodontic Archwire–Bracket Contacts

**DOI:** 10.3390/nano12193430

**Published:** 2022-09-30

**Authors:** Pengfei Wang, Xin Luo, Jiajie Qin, Zonglin Pan, Kai Zhou

**Affiliations:** Institute of Nanosurface Science and Engineering (INSE), Guangdong Provincial Key Laboratory of Micro/Nano Optomechatronics Engineering, College of Mechatronics and Control Engineering, Shenzhen University, Shenzhen 518060, China

**Keywords:** carbon film, graphene sheets, orthodontic archwire, bracket, fretting wear, artificial saliva

## Abstract

Carbon films were fabricated on the orthodontic stainless steel archwires by using a custom-designed electron cyclotron resonance (ECR) plasma sputtering deposition system under electron irradiation with the variation of substrate bias voltages from +5 V to +50 V. Graphene sheets embedded carbon (GSEC) films were fabricated at a higher substrate bias voltage. The fretting friction and wear behaviors of the carbon film-coated archwires running against stainless steel brackets were evaluated by a home-built reciprocating sliding tribometer in artificial saliva environment. Stable and low friction coefficients of less than 0.10 were obtained with the increase of the GSEC film thickness and the introduction of the parallel micro-groove texture on the bracket slot surfaces. Particularly, the GSEC film did not wear out on the archwire after sliding against three-row micro-groove textured bracket for 10,000 times fretting tests; not only low friction coefficient (0.05) but also low wear rate (0.11 × 10^−6^ mm^3^/Nm) of the GSEC film were achieved. The synergistic effects of the GSEC films deposited on the archwires and the micro-groove textures fabricated on the brackets contribute to the exceptional friction and wear behaviors of the archwire-bracket sliding contacts, suggesting great potential for the clinical orthodontic treatment applications.

## 1. Introduction

Currently, more and more people are selecting orthodontic treatment to correct their tooth position for pursuing both physical and mental health, and thus high quality of daily life [1,2]. A metallic orthodontic fixed appliance is composed of archwires and brackets, which has superior stabilization and could provide different types of orthodontic force, thus making it a popular product in clinical orthodontic treatment. Sliding friction coefficient, not only static friction coefficient, but also kinetic friction coefficient between the archwire and bracket during the tooth movement, is one of the most important issues in orthodontic treatment. The increase of the friction forces causes a direct decrease of the effective orthodontic force acting on the teeth, leading to pain and discomfort of the patients, damage of teeth, and related tissues, and finally prolonged treatment period [3,4,5]. Therefore, to accomplish the optimum design of low friction and light force orthodontic appliance, great efforts have been devoted for reducing both the static and kinetic friction coefficients in the archwire–bracket tribo-pairs in previous studies [6,7,8,9].

Surface modifications of either archwire or bracket have been clarified to be effective strategies for achieving low friction coefficients as well as low wear rates of archwire–bracket combinations in both dry and wet (artificial saliva or human saliva) conditions [10,11,12,13,14]. Particularly, several types of nano coatings have been produced for promoting the fiction and wear properties of the orthodontic archwire–bracket sliding contacts. Huang et al. [15] clarified that the deposition of diamond-like carbon (DLC) film on the nickel–titanium (NiTi) archwires could profoundly reduce the stable friction coefficient from 0.40 to 0.12 of the archwire–bracket sliding contacts in an artificial saliva environment. The deposition of carbon nitride (CNx) thin film on the surface of stainless steel archwire could definitely decrease the friction of the archwire–bracket in ambient air and artificial saliva environment [16,17,18]. The friction coefficients between the stainless steel orthodontic archwires and brackets were considerably decreased after the fabrication of chitosan (CTS) and zinc oxide (ZnO) coatings on both archwires and brackets [19]. The friction coefficients of the metal archwire–bracket tribo-pairs decreased from 0.552 to 0.207 and 0.372 by coating with Al_2_O_3_ and TiN, respectively [20].

For better reproducing the human oral cavity environment and mimicking the low-amplitude and long-duration movement of the archwire–bracket contact in the patient, an orthomicrotribometer was developed for characterizing the tribological properties of the orthodontic archwire–bracket tribo-pair by modulating the normal load, frequency, and environment during the sliding fretting wear tests. Moreover, a reduction of friction and wear was obtained with the increase of normal load and frequency [21]. Kang et al. [22] clarified the fretting wear properties of DLC films coated orthodontic archwires, and it was found that the friction and wear of the archwires were decreased with the fabrication of DLC films on the archwires.

In this work, we reported the fretting friction and wear behaviors of the carbon film-coated archwire produced by using an electron cyclotron resonance (ECR) plasma sputtering deposition system under different substrate bias voltages. The friction and wear properties of the archwire–bracket contact combinations were systematically investigated by using a reciprocating sliding archwire–bracket tribometer in artificial saliva environment. The low friction and low wear mechanisms of the graphene sheets embedded carbon (GSEC) film coated archwires in an artificial saliva environment were clarified.

## 2. Materials and Methods

Commercially available stainless steel orthodontic components, namely the archwires and brackets, were used in this research. The archwire (HXH-0023, Xihubiom, Hangzhou, China) with a rectangular cross-section of 0.017 inch × 0.025 inch (0.42 mm × 0.64 mm) and a straight length of 110 mm was fitted into the conventional stainless steel bracket (ALS-0008, Alice Dental, Hangzhou, China) with a slot size of 0.022 inch (0.56 mm). The archwires and brackets were ultrasonically cleaned in acetone, ethanol, and purified water in sequence for 20 min each to remove surface contaminations.

Carbon films were deposited onto the surfaces of the substrates by using a custom-designed ECR plasma sputtering deposition system under working mode of low-energy electron irradiation. The formation mechanisms of the graphene nano-sheets in the amorphous carbon matrix during the low-energy electron irradiation could be found in the previous works [23,24,25]. Four positive bias voltages of +5 V, +10 V, +20 V, and +50 V were applied on the substrates to produce carbon films with different types of nanostructures. The deposition time varied from 25 to 80 min for adjusting the thickness of the carbon films. Two types of substrates were employed for the preparation of carbon films. Specifically, the carbon films synthesized on the *p*-type (100) silicon substrates (dimensions of 20 mm × 20 mm × 0.525 mm) were used for the TEM–EELS analysis and nanoindentation test, whereas the carbon films fabricated onto the contact surface of stainless steel archwires (width of 0.017 inch) were applied for the fretting wear tests and corresponding characterizations.

The nanostructure and electric structure of the carbon films were examined by using a high resolution transmission electron microscope (HRTEM, Titan Cubed Themis G2 300, FEI, Hillsboro, OR, USA) in combination with an electron energy loss spectroscope (EELS, Quantum ER/965 P, Gatan, Pleasanton, CA, USA) working under an electron accelerating voltage of 80 kV to prevent damage or recrystallization of the specimens. The plan views of TEM specimens were obtained by scratching the surface of carbon film with a diamond pencil and then collecting the micro-flakes onto a copper grid. The bonding configuration information of the carbon films was measured via a Raman spectroscope (LabRAM HR Evolution, Horiba, Kyoto, Japan) with a laser wavelength of 532 nm. A laser power of 0.1 mW was employed for avoiding potential heating damage on the specimen surface. The hardness and Young’s modulus of the carbon films were evaluated from the nanoindenter (G200, Agilent, Santa Clara, CA, USA) with a Berkovich diamond indenter.

The fretting wear tests were conducted on a self-designed archwire–bracket reciprocating sliding tribometer, the experimental setup is shown in Figure 1. It mainly consisted of a reciprocating sliding system which drives the archwire horizontally while the mating bracket was vertically adjusted by a three-axis stage. As illustrated in Figure 1a,b, the archwire was installed on the archwire holder and pre-stretched with a tensile force of 10 N. The bracket holder was mounted at the front of the cantilever beam which had been attached to the three-axis stage. The loading and unloading processes of the contact combination of the archwire and bracket was controlled by manual manipulating the three-axis stage with a human hand. The reciprocating sliding between the stationary bracket and moving archwire was achieved by a stepper motor. The normal load and tangential friction forces during the fretting tests were recorded for calculating the friction coefficient.

As shown in Figure 1c, the archwire was slowly placed into the slot of the bracket during the loading process, and then the friction test started with a displacement stroke of 150 μm at a frequency of 0.5 Hz. The applied normal load was controlled between 0.5 N and 2.0 N. Each archwire–bracket tribo-pair was used only once. All the fretting tests were carried out in an artificial saliva environment with 10,000 s (10,000 times). Commercially available artificial saliva (pH = 6.9) with a precisely controlled temperature of 37 °C (simulation of the human oral environment) was supplied to the contact interfaces by using a tube with an inner diameter of 3.2 mm. The dripping speed of the artificial saliva was maintained at 3.6 mL/min with the help of a peristaltic pump. An optical microscope (Eclipse LV150N, Nikon, Tokyo, Japan) and a white-light interferometer (Contour GTX, Bruker, San Jose, CA, USA) was applied to observe the worn surfaces on the archwires and brackets and to estimate the fretting wear rate of the carbon films coated on the archwires as well. The element compositions of the worn surfaces on the carbon films coated archwires were analyzed by using a scanning electron microscope (SEM, Scios, FEI, Hillsboro, OR, USA) in combination with an energy dispersive X-ray spectroscope (EDS).

## 3. Results

### 3.1. Compositon and Structure Analysis of Carbon Films

The nanostructure and bonding configuration of the carbon films were analyzed by using TEM–EELS and Raman spectroscopy. The results are shown in Figure 2. Figure 2a–d indicate the plan view TEM images of the four types of carbon films fabricated on the silicon substrates with 25 min under different substrate bias voltages. The high resolution TEM images and dispersive rings in the fast Fourier transformation (FFT) images exhibited the formation of an amorphous carbon phase in the carbon films prepared under substrate bias voltages of +5 V and +10 V, as shown in Figure 2a,b. However, with the increase of the substrate bias voltage, a clear insertion of stacked layer structures into the amorphous structure with random orientation was observed on the TEM images of carbon films produced at substrate bias voltages of +20 V and +50 V, as shown in Figure 2c,d. The interplanar spacing was approximately 0.34 nm, which matched well with that of the graphene sheet. The normal size of the graphene sheets was several nanometers. A pair of white light spots was observed from the corresponding inset FFT images, which further confirmed the existence of the graphene sheets in the carbon films.

To confirm the bonding configuration of the carbon films, the EELS spectra and Raman spectra of the four types of carbon films were analyzed in Figure 2e,f. Figure 2e demonstrates the core loss EELS spectra of the carbon films after the subtraction of a power-law background. Specifically, two individual peaks at 285.0 eV (denoted as π* peak) and 293.0 eV (denoted as σ* peak) were observed on the EELS spectrum. The shape of the σ* peak was highly related to the carbon nanocrystalline structure. The transition from a round σ* peak to a sharp σ* peak was clearly observed on the EELS spectrum of the carbon films with the increase of the substrate bias voltage. The height ratio of π* peak to σ* peak (denoted as P1/P2) increased from 0.66 to 0.75 with the increase of the substrate bias voltage from +5 V to +50 V, respectively, as listed in Table 1. Hence, these observations suggested the formation of *sp*^2^ carbon nanocrystallites at a higher substrate bias voltage.

Figure 2f exhibits the Raman spectra (ranging between 1100 cm^−1^ and 3500 cm^−1^) of the four types of carbon films produced on the surface of the stainless steel archwires with 80 min under various substrate bias voltages. Clearly, at substrate bias voltages of +5 V and +10 V, there was no obvious 2D peak on the corresponding Raman spectrum, whereas the spectra of the carbon films prepared at +20 V and +50 V consisted of a distinct D peak at 1350 cm^−1^, a G peak at 1580 cm^−1^, and a well-defined 2D peak at 2700 cm^−1^. Generally, the appearance of two separated D and G peaks together with a distinct 2D peak in the spectrum strongly confirmed the formation of graphene nanocrystallites in the carbon film. The peak intensity ratio of D peak to G peak (denoted as *I*_D_/*I*_G_) was calculated by curve fitting of the D peak and G peak (1100 cm^−1^ to 2000 cm^−1^) with a Lorentzian function and a Breit–Fano–Wagner (BFW) function [25], respectively, after subtracting a linear background, the fitting curves are shown in Figure 2f. A continuous shift of D peak position to a lower value and G peak position to a higher value, together with a prominent increase of *I*_D_/*I*_G_ were observed with the increase of substrate bias voltages from +5 V to +50 V, as summarized in Table 1, which suggesting an increase of the *sp*^2^ carbon structure in the carbon films according to the Ferrari’s typical model [26]. The in-plane size (*L_α_*) of the nanocrystalline structure in the carbon film could be determined by using *I*_D_/*I*_G_ = *C*(λ)*L_α_*^2^, where *C*(λ) is closely related to the excited laser wavelength and equal to 0.55 nm^−^^2^ [27]. The calculated nano-size of the graphene nanocrystallite greatly increased from 1.13 nm to 1.74 nm with the increase of the substrate bias voltage from +5 V to +50 V, respectively. Therefore, it could be concluded that amorphous structure was generated in carbon film deposited at substrate bias voltages of +5 V and +10 V, whereas nanocrystalline graphene structure was formed in carbon films prepared at substrate bias voltages of +20 V and +50 V.

The hardness and Young’s modulus of the carbon film produced at silicon substrate with 80 min are also summarized in Table 1. It was found that the hardness drastically decreased from 12.1 to 0.8 GPa with the increase of substrate bias voltage from +5 V to +50 V. A similar tendency was observed for the Young’s modulus. These results agreed well with the increase of *sp*^2^ carbon bonding structures in the carbon films with increasing substrate bias voltage, as concluded from the TEM–EELS analysis.

### 3.2. Fretting Wear Behaviors of Carbon Film-Coated Archwires

Figure 3 illustrates the characteristic fretting wear tests of the carbon film-coated archwires sliding against conventional stainless steel brackets in an artificial saliva environment. The fretting wear test of the uncoated stainless steel archwire sliding against conventional stainless steel bracket was shown in Appendix A for comparison. High friction coefficients of 0.50 were observed for the contact combination of stainless steel archwire and bracket in artificial saliva environment. It was clearly shown that the friction coefficient of the archwire–bracket contact combination strongly decreased with the protection of carbon film on the archwire. As shown in Figure 3a–e, the friction coefficient first decreased at the initial running-in stage, followed by a steady stage with low friction coefficient of around 0.10. Thereafter, the friction coefficient increased and kept at a high value up to 0.30. It could be observed from the optical microscopy images of the wear scars on the archwires (shown in Figure 3f–i) that all the carbon films worn out on the archwires after 10,000 tests. Particularly, severe wear was observed on the wear scar of the carbon film prepared under the substrate bias voltage of +10 V.

To extend the lifetime of the carbon film on the stainless steel archwire from the aspect of clinical application, a series of carbon films were prepared at the substrate bias voltage of +50 V with a prolonged deposition time, the thickness of the deposited carbon film is shown in Appendix A, and the corresponding fretting wear results are shown in Figure 4. The friction curves showed a similar tendency for carbon film deposited at 25 min and 35 min, as shown in Figure 3d and Figure 4a, respectively. However, with the further increase of the deposition time, as shown in Figure 4b,c, the sharp increase of the friction coefficient disappeared at the latter half of the friction curves. Particularly, stable and low friction coefficients of approximately 0.07 were obtained after a quick running-in stage for carbon film deposited at 80 min, as shown in Figure 4d. Unfortunately, all the carbon films wore out on the wear scars of the archwires, as shown in Figure 4e–g. It could also be found that the peeling off area for carbon film deposited at 80 min was greatly minimized, as shown in Figure 4g. It was inferred that lower friction coefficient and lower wear rate of the carbon film could be obtained with the increased film thickness. Additionally, the effect of normal load (0.5 to 2 N) on the fretting wear behavior of the carbon film-coated archwire was studied, as shown in Appendix A. It could be seen that relative stable and low friction coefficients of less than 0.10 were obtained at normal load of 1.0 N and 1.5 N. However, the fabricated carbon films on the archwires wore out at all the normal load experimental conditions.

Great attention has been paid to improving the friction and wear properties of tribo-pairs with the elaborately prepared textured surfaces by laser surface texturing technique in the last few decades for fundamental research and industrial applications [28,29,30,31,32,33]. To promote the friction and wear behaviors of the carbon film-coated archwires, a laser surface texturing technique has been employed for manufacturing textures to effectively reduce both the friction and wear of the archwire–bracket sliding contacts in our previous work [34]. Specifically, three types of micro-groove textures were fabricated on the slot surfaces of the stainless steel brackets by using a picosecond laser processing apparatus (SLCUV-250-PS, Shengxiong, Dongguan, China). As shown in Figure 5, micro-groove texture with a width of 45 μm, a depth of 15 μm, and a row spacing of 150 μm was produced on the slot surfaces with one row, two rows, and three rows.

The fretting wear results of the carbon film (substrate bias voltage of +50 V and deposition time of 80 min, hereafter denoted as +50 V and 80 min) coated archwires sliding against brackets with micro-groove textured slot surfaces are shown in Figure 6. Obviously, the application of micro-groove texture on the bracket slot surface could significantly reduce the friction coefficient of the archwire–bracket contact combination. Moreover, the friction coefficient generally decreased with the increase of the groove number from one to three, as shown in Figure 6a–c. Particularly, stable and low friction coefficient of approximately 0.06 was achieved after 1000 s running-in process when the carbon film-coated archwire rubbing against three-row micro-groove textured bracket in an artificial saliva environment. The steady state average friction coefficient for the final 2000 s was calculated to be 0.05. Furthermore, the peel off area of the carbon film on the wear scar decreased with the increase of the groove number from one to three, as shown in Figure 6e–g. Most importantly, it was found that carbon film still covered the entire wear scar on the archwire when sliding against the three-row micro-groove bracket; a sliding contact combination of carbon film against stainless steel bracket was successfully realized in the fretting wear test; delamination of the carbon film did not occur on the surface of the stainless steel archwire. A mild wear of the GSEC film on the wear scar was definitely achieved, as shown in Figure 6g.

## 4. Discussion

### 4.1. SEM–EDX Mapping Analysis of the Wear Scars

In order to gain more insights into the low friction and low wear mechanisms of the GSEC film-coated stainless steel archwires in an artificial saliva environment, surface morphologies and element compositions of the wear scars on the GSEC film-coated archwires after fretting tests were analyzed by using SEM in combination with EDX mapping, as shown in Figure 7. Three typical wear scars were investigated, the first one was on the GSEC film (+50 V and 25 min) coated archwire after running against untextured bracket, the second one was on the GSEC film (+50 V and 80 min) coated archwire after running against untextured bracket, and the last one was on the GSEC film (+50 V and 80 min) coated archwire after running against three-row micro-groove textured bracket. Optical microscopy images of the three wear scars were also provided for comparison. Obviously, the GSEC films on the stainless steel archwires were severely destroyed after sliding against untextured stainless steel brackets, as shown in Figure 7a,f, even with the increase of the thickness of the GSEC film. The corresponding carbon element distributions on the wear scars (Figure 7c,h, vanish of carbon element in the wear scar area) further confirmed that the GSEC films were substantially peeled off from the archwires. However, with the introduction of three-row micro-groove texture on the bracket, delamination of the GSEC film could hardly be observed on the shallow wear scar (Figure 7k). A uniform distribution of carbon (C), oxygen (O), and ferrum (Fe) elements on the wear scar (Figure 7m–o) undoubtedly proved that the GSEC film did not wear out on the archwire, the entire wear scar was still fully covered by the GSEC film. The contact combination of the GSEC film rubbing against three-row micro-groove textured bracket was realized in the whole fretting wear tests in artificial saliva environment.

### 4.2. Specific Wear Rate of the GSEC Film-Coated Archwires

To further clarify the outstanding fretting wear performances of the GSEC film coated archwires sliding against micro-groove textured brackets, the wear scars on the GSEC film-coated stainless steel archwires were examined by using a white light interferometer, the corresponding results are shown in Figure 8. The cross-sectional profiles were obtained at five different locations along the length direction for each wear scar. The 3D images and 2D cross-sectional profiles of the wear scars on the archwires with other contact combinations are provided in Appendix A for reference. Typical 3D image and 2D cross-sectional profile of the wear scar on the GSEC film (+50 V and 25 min) coated archwire running against untextured bracket are shown in Figure 8a,c, respectively. The wear scars on the archwires were in the dimensions of approximately 150 μm (length) × 50 μm (width). The wear depth of the wear scars was around 1.0 μm, which was twice larger than the related film thickness (490 nm, as shown in Appendix A). On the other hand, with the fabrication of three-row micro-groove texture on the bracket, the length and width of the wear scar were 100 μm and 50 μm, respectively. Particularly, the maximum depth of the wear scar was significantly decreased to around 200 nm, which was much less than half of the film thickness, as shown in Figure 8b,d. These results again confirmed that the GSEC film did not wear out on the wear scar of the archwires after fretting tests.

An empirical method was used for determining the wear volume based on the assumption that the theoretical shape of the wear scar on the archwire was a semi-cylinder. Therefore, the wear volumes of the wear scars on the archwires after sliding against brackets for all the five different contact combinations were calculated from the related profile curves, as shown in Figure 8e. The corresponding wear rates of the GSEC film coated archwires were thus derived by using the classical Archard equation [35]:*V* = *wLd*,(1)
where *V* is the wear volume of the wear scar (mm^3^), *w* is the wear rate (mm^3^/Nm), *L* is the applied normal load (N), and *d* is the reciprocating sliding distance (m). As demonstrated in Figure 8f, the wear volume of the wear scar on the GSEC film (+50 V and 25 min) coated archwire was 72.65 × 10^−6^ mm^3^, and it greatly decreased to 41.96 × 10^−6^ mm^3^ with the increase of deposition time from 25 min to 80 min. With the introduction of the micro-groove textured bracket, the wear volume further decreased and the lowest wear volume of 0.17 × 10^−6^ mm^3^ was obtained with the three-row micro-groove bracket. Undoubtedly, the corresponding wear rate was strongly decreased from 4.84 × 10^−6^ mm^3^/Nm to 0.11 × 10^−6^ mm^3^/Nm.

### 4.3. Friction and Wear Mechanisms of the GSEC Film-Coated Archwires

The excellent low friction and low wear mechanisms of the GSEC film (+50 V and 80 min) coated stainless steel archwire sliding against three-row micro-groove textured stainless steel bracket in an artificial saliva environment are proposed and schematic illustrations are shown in Figure 9. On the one hand, the superior friction behaviors of the GSEC film-coated stainless steel archwires sliding against untextured stanines steel brackets in an artificial saliva environment have been ascribed to the formation of a homogeneous tribo-induced graphene-rich tribofilm after the initial running-in process in combination with the accumulation of the salivary adsorbed film at the steady stage on the contact interfaces [36], as also shown in Figure 9a,b. The orientation of the graphene sheets changed from vertical to horizontal during the friction process, leading to the lower shear strength of the sliding contact, as shown in Figure 9c,d. Hence, low friction coefficients in the order of 0.10 were substantially obtained after a running-in period with the deposition of GSEC film on the stainless steel archwires, as shown in Figure 3e. Moreover, with the increase of the GSEC film thickness, the low friction coefficients decreased to less than 0.10, as shown in Figure 4d. Furthermore, exceptional stable and low friction coefficients of around 0.05 were achieved with the utilization of micro-groove texture on the bracket, and the optimum result was obtained when using the three-row micro-groove textured bracket, as indicated in Figure 6d. On the other hand, low friction coefficients of less than 0.10 were observed after initial running-in processes with the GSEC film-coated archwires but failed subsequently (e.g., 1000 s). Namely, the GSEC film was worn out progressively and started to be destroyed after the running-in stage, which resulted in the exposure of stainless steel archwire substrate surface and thus relative higher friction coefficients (e.g., 0.25) were observed for the self-mated stainless steel sliding contact, as shown in Figure 3e and Figure 4d. 

Great efforts have been devoted to enhancing both the friction and wear performances of the solid lubrication film (e.g., diamond-like carbon (DLC) and layered lamellar materials) coated textured surfaces; synergistic effects of the fabricated lubrication film and surface textures facilitate the achievement of low friction coefficient and high wear resistance of the sliding contacts in dry and wet experimental conditions [37,38,39,40,41]. Therefore, it was assumed that the introduction of micro-groove textures on the bracket slot surfaces could be an effective strategy for retaining the stable and low friction coefficients during the entire fretting wear tests. Moreover, it was found that low friction coefficient of 0.05 and low wear rate of 0.11 × 10^−6^ mm^3^/Nm were both obtained when the GSEC film-coated archwire run against the three-row micro-groove textured bracket in the artificial saliva environment.

The characteristic advantages of the micro-groove textures could be summarized as follows. Firstly, wear debris generated after the delamination of the GSEC film undergoing abrasive wear at the contact interface and acted as a third body in the sliding contact of untextured bracket, which further aggravated the wear of the GSEC film on the wear scar of the archwire. The fabrication of the parallel micro-groove texture could entrap the wear particles detached from the GSEC film and flowed out of the contact interface together with the artificial saliva, as shown in Figure 9c,d, which was beneficial for minimizing the ploughing friction force and achieving the mild wear of the GSEC film on the contact interfaces. Secondly, the real contact area of the sliding tribo-pairs decreased with the increase of the row of the micro-groove texture, which reduced the number of effective micro-asperity contacts and avoided the severe wear of the GSEC film. Finally, the three-row micro-groove textures acted as the reservoirs for continuous storing and supplying of the solid lubrication materials (e.g., graphene-rich tribofilm) as well as artificial saliva on the contact interface to reduce both the friction and wear of the mating materials. In sum, it was argued that the combined effects of the GSEC film deposited on the archwire and micro-groove textures fabricated on the brackets resulted in the promising friction and wear behaviors of the stainless steel archwire–bracket sliding contacts, which demonstrates great potential for the clinical medical applications.

## 5. Conclusions

In this work, we reported the fretting friction and wear behaviors of the carbon film-coated orthodontic stainless steel archwires running against untextured and micro-groove textured stainless steel brackets in artificial saliva environment. Carbon films were deposited onto the surface of archwires with an advanced ECR plasma sputtering system working under low-energy electron irradiation mode with the substrate bias voltages varying from +5 V to +50 V. Graphene nanocrystallites were embedded into the carbon matrix at a higher substrate bias voltage from the TEM–EELS and Raman analyses. Stable and low friction coefficients of less than 0.10 were substantially achieved with the increase of the GSEC film thickness as well as the fabrication of the parallel micro-groove texture on the bracket slot surfaces. Particularly, the GSEC film did not wear out on the archwire after sliding against the three-row micro-groove textured bracket for 10,000 times fretting tests, not only low friction coefficient (0.05) but also low wear rate (0.11 × 10^−6^ mm^3^/Nm) of the GSEC film were obtained. The prominent low friction coefficient and high wear resistance of the archwire–bracket sliding contacts are attributed to the synergistic effects of the GSEC lubrication film deposited on the archwires and the parallel micro-groove textures fabricated on the bracket slot surfaces, which suggests great potential for clinical orthodontic treatment applications.

## Figures and Tables

**Figure 1 nanomaterials-12-03430-f001:**
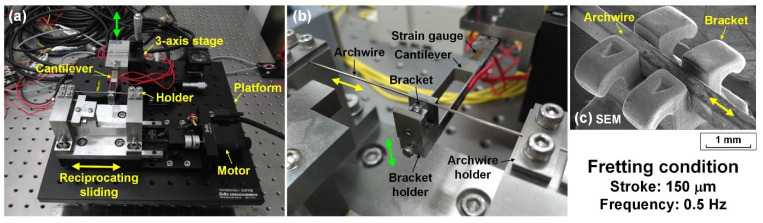
Experimental setup for the fretting wear tests. (**a**) Overview of the archwire–bracket reciprocating sliding tribometer. (**b**) Enlarged photo of contact combination of the stainless steel archwire and bracket. (**c**) SEM image of the contact combination of the stainless steel archwire and bracket.

**Figure 2 nanomaterials-12-03430-f002:**
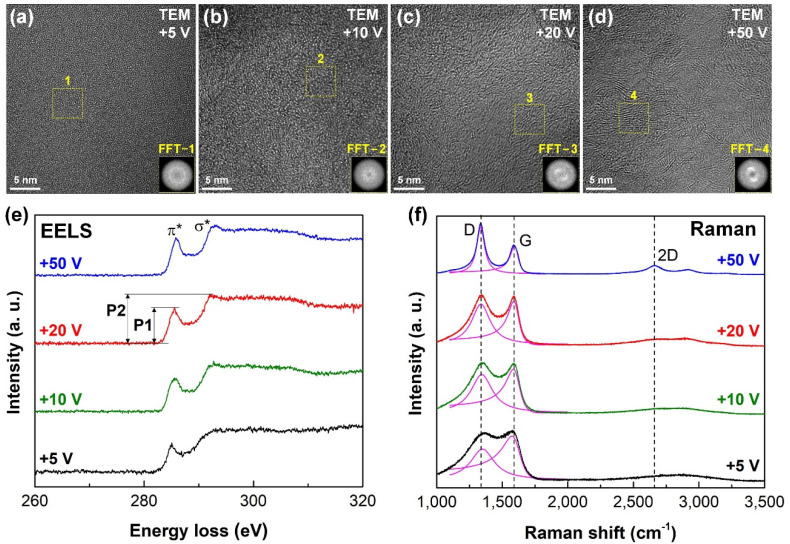
Characterization of the carbon films fabricated under different substrate bias voltages. HRTEM plan-view images of the carbon films deposited at substrate bias voltages of (**a**) +5 V, (**b**) +10 V, (**c**) +20 V, and (**d**) +50 V. Insets are the FFT images of the selected square region (marked with yellow dotted line). (**e**) EELS spectra and (**f**) Raman spectra of the carbon films.

**Figure 3 nanomaterials-12-03430-f003:**
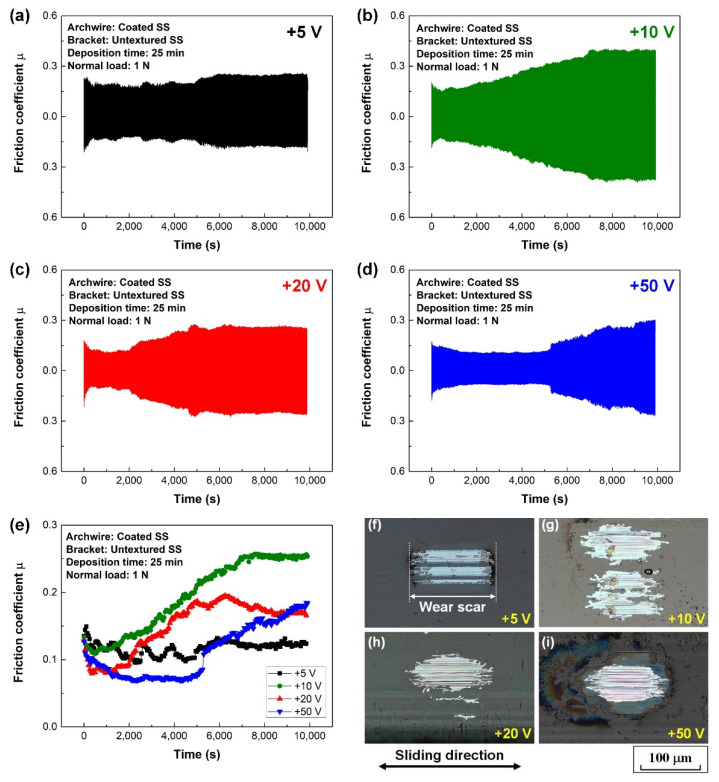
Fretting wear tests of carbon films coated stainless steel archwires running against conventional untextured brackets in artificial saliva environment. Friction curves of carbon films produced at substrate bias voltages of (**a**) +5 V, (**b**) +10 V, (**c**) +20 V, and (**d**) +50 V. (**e**) Average friction coefficients of different carbon films. Optical microscopy images of wear scars on the carbon films fabricated at substrate bias voltages of (**f**) +5 V, (**g**) +10 V, (**h**) +20 V, and (**i**) +50 V.

**Figure 4 nanomaterials-12-03430-f004:**
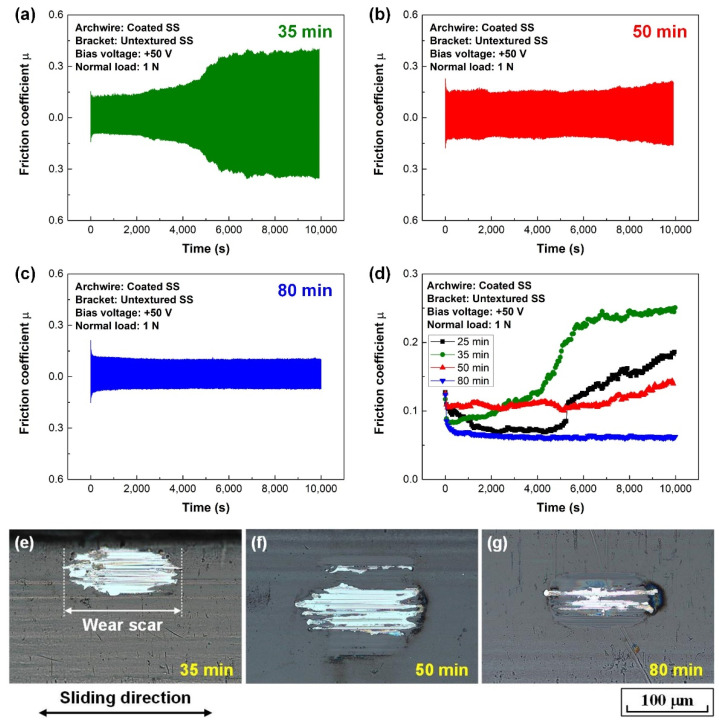
Fretting wear tests of different carbon film-coated stainless steel archwires running against conventional untextured brackets in artificial saliva environment. Friction curves of carbon films produced at the substrate bias voltage of +50 V and deposition times of (**a**) 35 min, (**b**) 50 min, and (**c**) 80 min. (**d**) Average friction coefficients of different carbon films. Optical microscopy images of the wear scars on carbon films with deposition times of (**e**) 35 min, (**f**) 50 min, and (**g**) 80 min.

**Figure 5 nanomaterials-12-03430-f005:**
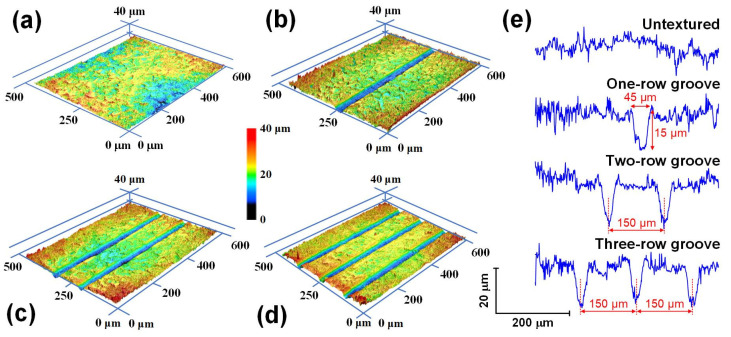
Surface morphologies of bracket slots with different textures. Three-dimensional optical images of bracket with (**a**) no texture, (**b**) one-row groove, (**c**) two-row groove, and (**d**) three-row groove. (**e**) Two-dimensional cross-sectional profiles of untextured and micro-groove textured stainless steel brackets.

**Figure 6 nanomaterials-12-03430-f006:**
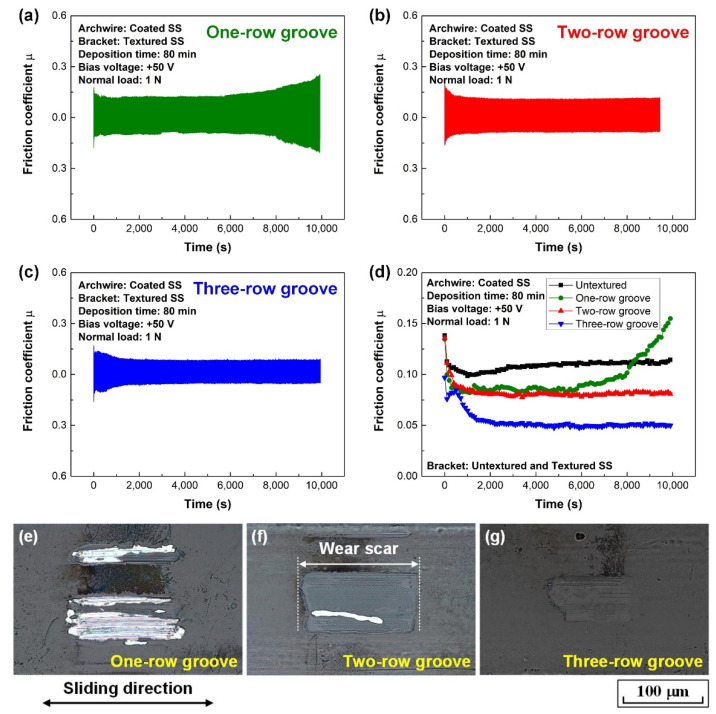
Fretting wear tests of carbon film (+50 V and 80 min) coated stainless steel archwires running against micro-groove textured brackets in artificial saliva environment. Friction curves of textured bracket with (**a**) one-row groove, (**b**) two-row groove, and (**c**) three-row groove. (**d**) Average friction coefficients of different textured brackets. Friction curve of untextured bracket is also shown for reference. Optical microscopy images of wear scars on the carbon films after sliding against brackets with (**e**) one-row groove, (**f**) two-row groove, and (**g**) three-row groove.

**Figure 7 nanomaterials-12-03430-f007:**
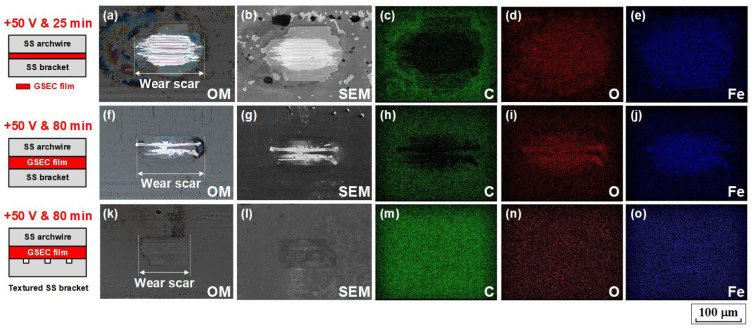
Surface characterizations of the wear scars on the GSEC film-coated stainless steel (SS) archwires after sliding against untextured and three-row groove textured stainless steel brackets in artificial saliva environment. (**a**–**e**) Wear scar on the GSEC film (+50 V and 25 min) coated archwire after running against untextured bracket. (**f**–**j**) Wear scar on the GSEC film (+50 V and 80 min) coated archwire after running against untextured bracket. (**k**–**o**) Wear scar on the GSEC film (+50 V and 80 min) coated archwire after running against three-row micro-groove textured bracket. (**a**,**f**,**k**) Optical microscopy images and (**b**,**g**,**l**) SEM images of the wear scars on the archwires. (**c**,**h**,**m**) carbon element, (**d**,**i**,**n**) oxygen element, and (**e**,**j**,**o**) ferrum element distributions of the wear scars on the archwires. Illustrations of three contact models are also shown for reference.

**Figure 8 nanomaterials-12-03430-f008:**
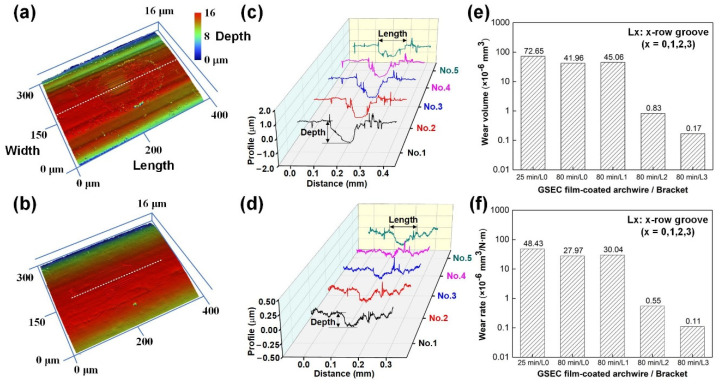
Calculation of the specific wear rate of GSEC film-coated archwires after sliding against untextured and micro-groove textured brackets for 5000 fretting cycles (10,000 times) in artificial saliva environment. (**a**) 3D image and (**c**) 2D cross-sectional profile of the wear scar on the GSEC film (+50 V and 25 min) coated archwires after running against untextured (denoted as L0) bracket. (**b**) 3D image and (**d**) 2D cross-sectional profile of the wear scar on the GSEC film (+80 V and 25 min) coated archwire after running against three-row micro-groove (denoted as L3) textured bracket. Cross-sectional profiles were obtained at five different positions along the length direction (white dotted line) of the wear scar. Calculated (**e**) wear volume and (**f**) wear rate of the GSEC film-coated stainless steel archwires. The Y axes are in log scale.

**Figure 9 nanomaterials-12-03430-f009:**
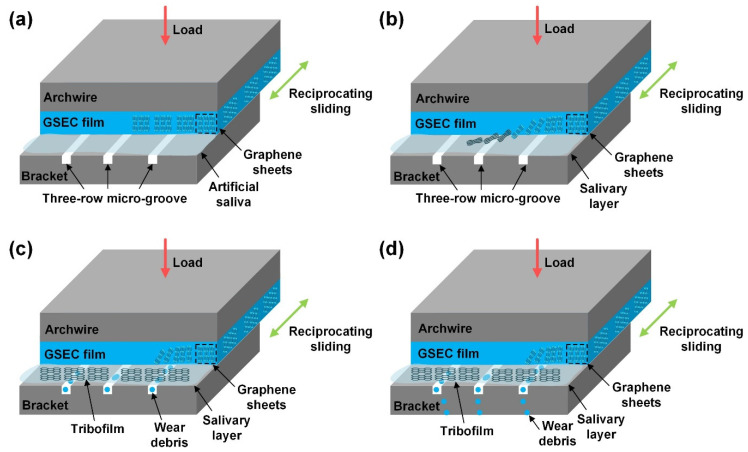
The low friction and low wear mechanisms of the GSEC film (+50 V and 80 min) coated stainless steel archwires running against three-row micro-groove textured stainless steel brackets in an artificial saliva environment. (**a**) Initial state of the contact combination, (**b**) stable low friction with the formation of graphene-rich tribofilm and salivary adsorbed layer, (**c**) accumulation of wear debris detached from the GSEC film with micro-groove, and (**d**) flow out of wear debris with artificial saliva from the micro-groove.

**Table 1 nanomaterials-12-03430-t001:** EELS, Raman, and mechanical parameters of the carbon films produced at different substrate bias voltages.

Substrate Bias Voltage (V)	EELS P_1_/P_2_	D Peak Position (cm^−1^)	G Peak Position (cm^−1^)	*I*_D_/*I*_G_	*Lα* (nm)	Hardness(GPa)	Young’s Modulus (GPa)
+5	0.66	1349	1574	0.70	1.13	12.1	123.0
+10	0.67	1344	1586	0.88	1.26	4.6	35.4
+20	0.72	1338	1589	0.94	1.31	4.0	31.5
+50	0.75	1336	1594	1.66	1.74	0.8	7.1

## Data Availability

The data that support the findings of this study are available from the corresponding author upon reasonable request.

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
