# Peer review of "Effect of Graphene Sheets Embedded Carbon Films on the Fretting Wear Behaviors of Orthodontic Archwire–Bracket Contacts"

_nanomaterials, 2022, doi:10.3390/nano12193430_

Round 1

Reviewer 1 Report

Authors attempted to study the graphene-coated dental wire and bracket contacts, but it requires addressing the following major observations:

1) Since coating thickness is only a few hundred nanometers; so within no time of tribology studies, the coated layer will wear out. As a result, whatever, studies reported in the present study seem to be the substrate; not the coating; as is clearly evident from the SEM images of worn surfaces.

2) Fig. 5: also indicates the above observations of the worn surface are a few tens of microns; supporting the above comment.

3) Presented COF flats are not acceptable; this form of data completely misled the tribology scientific community. really didn't understand the negative COF values. It seems to be raw flats; need to be presented in a proper way: can refer to these: https://doi.org/10.1016/j.ijrmhm.2021.105752; or https://doi.org/10.1016/j.ijrmhm.2019.105055

Reviewer 2 Report

Authors present a microscale tribological characterization of an Orthodontic Archwire-Bracket Contacts under realistic conditions by fretting tribology measurements. A number of microscopy and spectroscopy techniques are used to characterize the contact region, the wears tracks and the mechanisms at play. Carbon Coating on archwire as well texturing on the Bracket were realized by the authors.

The sample growth and its characterization, and in general the measurement procedures are very well done and all details are presented in the text very clearly with a correct and proper use of the English language.

 The optimal solutions proposed by the Authors are very well explained and reproducible.

For all that reasons, to my opinion, the paper could be published without significant modifications.

 I have only one concerns and that is whether this type of experiment, even if very well-constructed and well-conducted nevertheless exploring macro or eventually micron scale phenomena, is appropriate for the Nanomaterials journal

Round 2

Reviewer 1 Report

Authors addressed all my queries and suggestions; hence this manuscript may be accepted in the present format